# Association of the Estimated Pulse Wave Velocity with Cardio-Vascular Disease Outcomes among Men and Women Aged 40–69 Years in the Korean Population: An 18-Year Follow-Up Report on the Ansung–Ansan Cohort in the Korean Genome Environment Study

**DOI:** 10.3390/jpm12101611

**Published:** 2022-09-30

**Authors:** Byung Sik Kim, Yonggu Lee, Jin-Kyu Park, Young-Hyo Lim, Jeong-Hun Shin

**Affiliations:** 1Division of Cardiology, Department of Internal Medicine, Hanyang University Guri Hospital, Hanyang University College of Medicine, Guri-si 11923, Korea; 2Division of Cardiology, Department of Internal Medicine, Hanyang University Seoul Hospital, Hanyang University College of Medicine, Seoul 04763, Korea

**Keywords:** pulse wave velocity, cardiovascular mortality, cardiovascular event, general population

## Abstract

The estimated pulse wave velocity (ePWV) can predict adverse cardiovascular disease (CVD) outcomes in patients with increased CVD risks. However, data on its predictive capacity for CVD outcomes in the general population are limited. This study aimed to investigate the association between the ePWV and CVD outcomes among Korean adults. Ten thousand thirty patients aged 40–69 years from the Ansung–Ansan cohort in a prospective community-based cohort study were followed up for over 18 years. The ePWV was categorized into quartiles. Cox proportional hazard models were used to estimate the risk of cardiovascular (CV) mortality and CVD outcomes (composites of CV mortality, myocardial infarction, coronary artery disease, stroke, heart failure, and peripheral artery disease). The incidence of CV mortality and CVD outcomes was 7.0% and 22.1% in the fourth (highest) ePWV quartile and 0.1% and 4.5% in the first (lowest) quartile, respectively. After relevant covariate adjustments, the patients in the fourth quartile showed a significantly higher CV mortality risk (hazard ratio (HR), 7.57; 95% confidence interval (CI), 1.83–31.25). The patients in the third and fourth quartiles had higher CVD outcome risks (third: HR, 1.61; 95% CI, 1.19–2.16; fourth: HR, 1.56; 95% CI, 1.05–2.31) than those in the first quartile. This association was more clearly observed among women than among men. An elevated ePWV is associated with CV mortality and CVD outcomes. The ePWV is expected to serve as a potential marker for identifying high-risk groups for CVD events.

## 1. Introduction

Aortic stiffness refers to the elasticity of the blood vessel wall, and elevated aortic stiffness may result from and contribute to increased stress on the vessel walls [1]. Several studies have documented the prognostic importance of aortic stiffness as an independent predictor of cardiovascular (CV) mortality and all-cause mortality [2,3]. The carotid–femoral pulse wave velocity (cfPWV) is the gold standard parameter for measuring large arterial stiffness because of the relative ease of determination, perceived reliability, and, most importantly, prognostic implication demonstrated by a large body of evidence [2,3,4,5]. Moreover, the additive value of the cfPWV above and beyond traditional risk factors, including Systematic Coronary Risk Evaluation (SCORE) system findings and the Framingham risk score, has been suggested by several studies [6].

The estimated pulse wave velocity (ePWV), which is calculated from age and mean blood pressure (MBP) using an equation generated from the Reference Values for Arterial Stiffness Collaboration, has been reported to have a predictive value similar to that of the cfPWV [4,7]. Moreover, recent studies have shown that it can predict adverse cardiovascular disease (CVD) outcomes independent of traditional CVD risk factors in patients at increased risk of CVD [7,8,9]. However, its predictive capacity for CVD events in the general population has not yet been well established. Therefore, this study aimed to investigate the association between the ePWV and CVD outcomes in middle-aged Korean adults.

## 2. Materials and Methods

### 2.1. Study Participants

We analyzed data from the Ansung–Ansan cohort, which was a prospective study population consisting of 10,030 South Koreans aged 40–69 years residing in two cities, Ansung and Ansan. This cohort study started in 2001 and was embedded within the Korean Genome and Epidemiology Study (KoGES), a population-based cohort study funded by the Korean government, to investigate the genetic and environmental aetiologies of the prevalent metabolic and CVDs in South Korea. Detailed information regarding the study protocol has been reported previously [10]. Briefly, comprehensive health examinations, on-site interviews, and laboratory tests were conducted at each visit to a tertiary hospital located in the region. Nine serial assessments that completed the entire cohort protocol were performed after the baseline assessment through scheduled biennial revisits to the hospital until 2020. This study excluded individuals who had established CVD at baseline, thus, 9698 participants were finally included.

### 2.2. Consent

All participants voluntarily enrolled in the study, and written informed consent was obtained from all of them. The study protocol adhered to the principles of the Declaration of Helsinki and was approved by the Korean National Research Institute of Health and the Institutional Review Board (IRB) of Hanyang University Guri Hospital (IRB No. GURI-2021-09-016).

### 2.3. Lifestyle, Physical Activity, and Medical History Assessment and Physical Examination

Information on smoking, alcohol intake, education, income, marital status, and the presence of medical conditions, including hypertension, diabetes mellitus, dyslipidemia, myocardial infarction (MI), coronary artery disease (CAD), heart failure (HF), peripheral artery disease (PAD), ischemic and hemorrhagic stroke, and chronic kidney disease (CKD), was obtained using a questionnaire administered by trained investigators at the tertiary hospital at every visit. Data on the presence of regular exercise activity, type of exercise, weekly exercise frequencies and durations, daily physical activities, and duration of physical activities were also obtained using the questionnaire. The total physical activity per week was calculated as the summation of the metabolic equivalent task (MET) score of exercise activities per week and routine physical activities per week. Hypertension was defined as systolic blood pressure (SBP) or diastolic blood pressure (DBP) of ≥140 or ≥90 mmHg, respectively, or use of antihypertensive medications [11]. Diabetes mellitus was defined as either a fasting blood glucose level of ≥126 mg/dL, a hemoglobin A1c level of ≥6.5%, or use of medications for diabetes mellitus [12]. Dyslipidaemia was defined as a total cholesterol level of ≥240 mg/dL, low-density lipoprotein (LDL) cholesterol level of ≥160 mg/dL, triglyceride level of ≥200 mg/dL, high-density lipoprotein (HDL) cholesterol level of <40 mg/dL, or use of lipid-lowering medications [13].

Blood pressure (BP) was measured using a mercury sphygmomanometer by trained examiners at least two times at the level of the heart in a sitting position and was averaged. When there was a BP difference of ≥5 mmHg between the two measurements, a third measurement was obtained, and the last two measurements were averaged. Blood samples were collected after overnight fasting and analyzed using an automated analyzer (Hitachi Automatic Analyzer 7600, Hitachi, Nittobo, Tokyo, Japan). The estimated glomerular filtration rate (eGFR) was calculated using the Modification of Diet in Renal Disease (MDRD) equation [14].

### 2.4. Outcome Definition

CV mortality was identified using the Korean National Database for the Causes of Death registered in the Korean National Statistics Office. The database records the causes of death using the International Classification of Diseases-10 (ICD-10) codes. CV mortality was defined as ICD-10 codes I20–I82 (including ischemic heart diseases, HF, ventricular arrhythmia, ischemic and hemorrhagic stroke, and pulmonary thromboembolism). Newly developed MI, CAD other than MI, ischemic and hemorrhagic stroke, HF, and PAD were identified during the on-site interviews using a questionnaire at every visit. MI was defined as an urgent clinical event recalled by a participant as an MI requiring hospitalization or revascularization. CAD other than MI was defined by excluding MI among clinical events recalled by a participant as coronary artery disease requiring hospitalization or revascularization. HF was defined as a clinical event recalled by a participant as requiring hospitalization. PAD was defined as a clinical event recalled by a participant as requiring revascularization. Stroke was defined as an urgent clinical event recalled by a participant as stroke, sudden paralysis, or speaking difficulties requiring hospitalization. CVD outcomes were defined as a composite of CV mortality, MI, CAD other than MI, ischemic and hemorrhagic stroke, HF, and PAD.

### 2.5. ePWV Calculation

As described by Greve et al. [7], we calculated the ePWV for individuals with cardiovascular risk factors using Equation (1).
ePWV = 9.58748315543126 − 0.402467539733184 × age + 4.56020798207263 × 10^−3^ × age^2^ − 2.6207705511664 × 10^−5^ × age^2^ × MBP + 3.1762450559276 × 10^−3^ × age × MBP − 1.83215068503821 × 10^−2^ × MBP(1)

We calculated the ePWV for those without cardiovascular risk factors using Equation (2).
ePWV = 4.62 − 0.13 × age + 0.0018 × age^2^ + 0.0006 × age × MBP + 0.0284 × MBP(2)

The MBP was calculated as DBP + 0.4 × (SBP − DBP). The individuals without cardiovascular risk factors were defined as nonsmokers without any components of metabolic syndrome and without a history of MI or stroke. The components of metabolic syndrome were as follows [15]: (1) Abdominal obesity, defined as a waist circumference of ≥90 cm for men or ≥85 cm for women (following Korean-specific cut-offs for abdominal obesity defined by the Korean Society of Obesity) [16], (2) hypertriglyceridemia, defined as a serum triglyceride level of ≥150 mg/dL or specific treatment for this lipid abnormality, (3) low HDL cholesterol level, defined as a serum HDL cholesterol level of <40 mg/dL for men or <50 mg/dL for women or specific treatment for this lipid abnormality, (4) high BP, defined as an SBP of ≥130 mmHg and a DBP of ≥85 mmHg or treatment with antihypertensive agents, and (5) high fasting glucose level, defined as a fasting serum glucose level of ≥100 mg/dL or current use of antidiabetic medications. The ePWV was categorized according to quartiles (<7.39 m/s (first quartile, lowest), 7.39–8.44 m/s (second quartile), 8.45–9.89 m/s (third quartile), and >9.89 m/s (fourth quartile, highest)).

### 2.6. Statistical Analysis

Continuous variables were compared using one-way analysis of variance with Tukey’s post hoc test. Categorical variables were compared using the chi-square test. Continuous variables with skewed distributions were compared using the Kruskal–Wallis test. Kaplan–Meier survival analysis with log-rank test was used to compare the cumulative incidence of CV mortality and CVD outcomes among the groups. We used a multivariable Cox proportional hazard model to investigate the association between the ePWV and CV mortality and CVD outcomes with or without adjustment for the selected confounders. We used an unadjusted model and three different adjusted models. In model 1, age, sex, and SBP were considered possible confounders. Model 2 included clinically relevant lifestyle factors: Age, sex, smoking status, alcohol consumption status, physical activity, income level, and educational status. Model 3 included the variables in Model 2 and clinically relevant medical factors: Variables included in Model 2 plus body mass index, waist circumference, medical history (hypertension, diabetes mellitus, dyslipidemia), eGFR, fasting glucose level, total cholesterol level, and LDL cholesterol level (per 1 mg/dL). Hazard ratios (HRs) and 95% confidence intervals (CIs) were calculated. Additionally, we performed a subgroup analysis according to sex with multivariable Cox regression analysis using the same adjusted variables as in Model 3.

We also examined whether the addition of the ePWV was associated with an improvement in the prediction model of CV mortality and CVD outcomes with the 10-year atherosclerotic cardiovascular disease (ASCVD) risk score [17]. Harrell’s C-index was used to measure the discriminative ability across the models and was compared using a method introduced by Haibe-Kains et al. [18]. We also used the net reclassification index (NRI) to evaluate the improvements in the predictive performance. For the purpose of finding an optimal cut-off level of ePWV for predicting CVD events, we used time-dependent receiver operating characteristic (ROC) curve analysis. Time-dependent ROC curve analysis was estimated using sensitivity and 1—specificity, which are obtained from various cut-off levels of ePWV at specific time points. The Youden index method was used to calculate optimal cut-off levels. All statistical analyses were conducted using the open-source statistical software R (version 4.1.0, www.R-project.org, accessed on 27 May 2021) and R-studio (version 1.4.1, www.rstudio.com, accessed on 27 May 2021) and statistical packages, including rms, descr, survival, tableone, survminer, survcomp, timeROC, and nricens. A *p*-value of <0.05 was considered statistically significant.

## 3. Results

### 3.1. Baseline Characteristics

A total of 9698 participants (men, 47.2%; mean age, 52.1 [SD, 8.9] years) were analyzed. The median follow-up period was 16.8 (interquartile range, 9.0–16.9) years. The baseline characteristics according to the quartiles of the ePWV are shown in Table 1. The higher the quartile of the ePWV, the higher the mean age, waist circumference, SBP, DBP, eGFR, and levels of fasting blood glucose, hemoglobin A1c, and total cholesterol. The higher the quartile of the ePWV, the higher the frequencies of low-income level and low educational status. In addition, the frequency of medical histories of hypertension, diabetes mellitus, and CKD increased with a higher quartile of the ePWV. In contrast, higher frequencies of current smoking and decreased physical activity were observed in the lower quartile of the ePWV.

### 3.2. ePWV and Cardiovascular Events

During the observation period, 217 CV mortality and 1219 CVD outcomes were observed. The cumulative incidence of CV mortality and CVD outcomes increased with a higher quartile of the ePWV (Figure 1). To examine the independent association between the ePWV and CV mortality and CVD outcomes, we performed Cox regression analysis (Table 2 and Appendix A). The univariate Cox proportional hazard models showed that the risk of CV mortality and CVD outcomes increased with a higher quartile of the ePWV. After adjustments for relevant variables, the patients in the fourth (highest) quartile of the ePWV (HR, 7.57; 95% CI, 1.83–31.25) showed a significantly higher risk of CV mortality than did those in the first (lowest) quartile of the ePWV. In addition, the patients in the third (HR, 1.61; 95% CI, 1.19–2.16) and fourth quartiles of the ePWV (HR, 1.56; 95% CI, 1.05–2.31) showed a significantly higher risk of CVD outcomes than did those in the first quartile of the ePWV. The optimal cut-off levels of ePWV for discrimination between patients with and without CVD events obtained from time-dependent ROC curve analysis at 120, 144, 168, 192, and 216 months are shown in Appendix A. The optimal cut-off levels of ePWV were estimated to be between 8.82 and 10.08 m/s.

### 3.3. ePWV and Cardiovascular Events According to Sex

We also performed a subgroup analysis stratified by sex. The baseline characteristics according to sex are shown in Appendix A. The men were younger (mean age: 51.6 versus 52.5 years; *p* < 0.001) and had higher SBP and DBP than the women. The men were also significantly more likely to be current smokers and alcohol drinkers than the women. In addition, the frequency of diabetes mellitus and dyslipidemia was higher among men than among women. Conversely, the frequency of hypertension and CKD was higher among women than among men. Figure 2 shows the association between the ePWV and CV mortality and CVD outcomes according to sex. In the univariate Cox proportional hazard models, the risk of CV mortality and CVD outcomes increased with a higher quartile of the ePWV in both women and men. After adjustments for relevant variables, there were no significant differences in the risk of CV mortality and CVD outcomes according to the ePWV quartiles among the men. However, the women in the fourth quartile of the ePWV showed a significantly higher risk of CV mortality than the women in the first quartile of the ePWV. In addition, the patients in the second, third, and fourth quartiles of the ePWV showed a significantly higher risk of CVD outcomes than the patients in the first quartile of the ePWV.

### 3.4. Incremental Value of the ePWV in Predicting Cardiovascular Events

We performed another analysis to evaluate whether adding the ePWV to the well-validated risk prediction model (10-year ASCVD risk score) has additional value in predicting CV mortality and CVD outcomes. A comparison between Cox proportional hazard models for the prediction of CV mortality and CVD outcomes is shown in Table 3. The Harrell’s C-index for CV mortality and CVD outcomes in relation to the 10-year ASCVD risk score were 0.809 (95% CI, 0.784–0.835) and 0.687 (95% CI, 0.673–0.701); ePWV, 0.810 (95% CI, 0.784–0.835) and 0.684 (95% CI, 0.669–0.698); and combined model, 0.824 (95% CI, 0.801–0.849) and 0.697 (95% CI, 0.683–0.711). Adding the ePWV to the Cox proportional hazard models with the 10-year ASCVD risk score modestly increases the C-index (*p* = 0.023 for CV mortality and *p* = 0.008 for CVD outcomes). The Bayesian information criterion (BIC) of the Cox proportional hazard models also improved after ePWV was added to the model with the 10-year ASCVD risk score (ΔBIC = 88.9 for CV mortality and ΔBIC = 139.9 for CVD outcomes). In addition, overall NRI showed slight improvements in classification ability of the Cox proportional hazard models for CV mortality (NRI, 0.094; 95% CI, 0.032–0.162) and the CVD outcomes (NRI, 0.105; 95% CI, 0.054–0.172), when the ePWV was added to the 10-year ASCVD risk score (Appendix A).

## 4. Discussion

This study investigated the association between the ePWV and the risk of CVD events in middle-aged Korean adults. The participants in the higher quartiles of the ePWV had a higher risk of CV mortality and CVD outcomes independent of other confounding variables than their counterparts. In particular, the association of the ePWV with an increased risk of CV mortality and CVD outcomes was observed more clearly in the women than in the men. Furthermore, the addition of the ePWV to the Cox proportional hazard models with the 10-year ASCVD risk score modestly improves the discrimination between CV mortality and CVD outcomes. Our study results imply that the ePWV, which reflects aortic stiffness, may be a simple and useful parameter for predicting CVD events in middle-aged Koreans.

Aortic stiffness is an indicator of subclinical disease and is associated with an increased risk of various diseases, such as hypertension, CKD, and stroke [19]. Furthermore, increased aortic stiffness has been associated with an increased risk of CVD events in the general population as well as in high-risk populations, such as those with hypertension, diabetes mellitus, CKD, and CAD [3,20,21,22,23]. Given the accumulating evidence, including the aforementioned findings, the 2020 International Society of Hypertension guideline for the management of hypertension and 2021 European guidelines for CVD prevention in clinical practice recommend the cfPWV as a parameter for assessing arterial hypertension-mediated organ damage [24,25]. However, its widespread use is controversial owing to measurement difficulties and substantial publication bias [25,26]. Therefore, there is a clinical need for an easy and relevant marker of aortic stiffness that can replace the cfPWV [27].

Recently, Greve et al., 2016 proposed a new, simple parameter for predicting the cfPWV, called the ePWV, based on the determination of reference values for the cfPWV [4,7]. They demonstrated that the ePWV had a predictive value for future CVD events, especially in healthy individuals and those with untreated hypertension [7,28]. More recently, the same group reported that the ePWV was an independent risk factor for CVD events in healthy individuals by analyzing a cohort consisting of a larger number of healthy individuals [29]. Other studies have also reported that the ePWV is associated with CAD [30] and stroke [31]. Furthermore, Vlachopoulos et al., 2019 found that the ePWV predicted CVD events in the Systolic Blood Pressure Intervention Trial (SPRINT). They also suggested that individuals whose ePWV responded to antihypertensive treatment showed a lower risk of all-cause death than did non-responders, independent of changes in the SBP [8]. In agreement with previous reports mainly from Western populations, the results of our study on the Korean general population added evidence that as a marker of aortic stiffness, the ePWV had prognostic implications for the future risks of CV mortality and CVD outcomes, defined as a composite of CV mortality, MI, CAD other than MI, ischemic and hemorrhagic stroke, HF, and PAD.

Previously, Jae et al., 2020 reported that the ePWV was significantly associated with CVD events in men [9]. They analyzed a cohort exclusively comprising men and showed that the ePWV was a risk factor for all-cause death, CV mortality, sudden cardiac death, and atrial fibrillation in men. Conversely, in our study, the ePWV was not associated with an increased risk of CV mortality and CVD outcomes among men but was significantly associated with an increased risk among women. These conflicting results may be attributed to the demographic differences in the study participants. In our study, the men had relatively higher frequencies of traditional cardiovascular risk factors, such as smoking status, diabetes mellitus, and dyslipidemia, than the women. Therefore, it can be inferred that the men were affected by the high burden of these cardiovascular risk factors. Therefore, the effect of the ePWV would be small, and the prognostic effect was more prominent among the women who had a relatively low CVD risk. Further studies in different populations, racial/ethnic backgrounds, and various clinical situations are warranted. Moreover, the association strength of ePWV and CVD events in both sexes decreased as more variables were adjusted. This may be because ePWV is calculated with age and MBP. The majority of covariates included in the Cox proportional hazard models are variables that can change as increased age or BP. In addition, age and SBP are also included as covariates. Therefore, association strength seems to decrease in the multivariable model. However, even in a fully adjusted model, ePWV was still significantly associated with CVD events in women. A large prospective cohort study is needed to clarify the role of ePWV in predicting CVD events. One interesting issue regarding the ePWV is whether it has an incremental value in predicting CVD events when added to conventional well-validated predictive models. In the SPRINT study population, which consisted of patients with hypertension, adding the ePWV slightly improved the C-index for the primary outcome from 0.676 to 0.683 (*p* = 0.049) and improved the C-index for all-cause death from 0.67 to 0.69 (*p* = 0.03). The NRI for survival compared with the Framingham risk score was 0.111 (*p* < 0.001) [8]. Vishram-Nielsen et al., 2020 concluded that adding the ePWV to the Cox model with SCORE or Framingham risk score did significantly increase the area under the ROC curves, but not in the Cox model with conventional risk factors [29]. In our study, adding the ePWV to the Cox proportional hazard models with the 10-year ASCVD risk score did significantly improve the C-index for CV mortality and CVD outcomes. In addition, an improvement in the overall NRI was also observed. Therefore, our results may support the rationale for using the ePWV in conjunction with the conventional risk model for the prediction of CVD events. Collectively, considering that the ePWV is a risk factor for CVD events independent of confounding variables and can be obtained with simple indicators, such as age and BP, it could be used as a simple marker to identify high-risk groups for CVD events in clinical practice.

The strength of our study is the large sample size and the 18-year follow-up period. Another strength was the inclusion of a community-based general population commonly encountered in clinical practice, rather than a specific population group. In addition, the quality of this study was enhanced by conducting face-to-face interviews at each examination, in strict observance of the standardized protocol.

This study has several limitations. Firstly, this was an observational study. Therefore, any association between the ePWV and outcomes cannot be directly interpreted as a causal relationship. Despite adjustments for various confounding factors, there is a possibility of unmeasured confounding factors. Secondly, the study population comprised residents of two cities in Korea, suggesting the possibility of selection bias. Thirdly, information about physical activity, previous medical history, and medication history was obtained using a questionnaire. Therefore, recall bias might have been present. In addition, detailed medication lists were lacking. Further studies are needed to evaluate the effect of a specific disease or treatment on ePWV. Finally, during the 18-year follow-up period, 38.6% of the participants dropped out from the follow-up assessments, which might have resulted from serious illnesses, such as CVD events. Therefore, the number of cardiovascular outcomes, except for CV mortality, might have been underestimated. However, since CV mortality was identified using the Korean National Database, it was also accurately reflected among the dropped-out participants.

## 5. Conclusions

This study demonstrated that the ePWV was an independent risk factor for CVD events. Moreover, the addition of the ePWV significantly improves the conventional prediction model for CVD events. Therefore, ePWV may be a useful marker that can be easily used without specific equipment in clinical practice. Further studies are needed to determine a marker of aortic stiffness that can replace the cfPWV in various clinical situations.

## Figures and Tables

**Figure 1 jpm-12-01611-f001:**
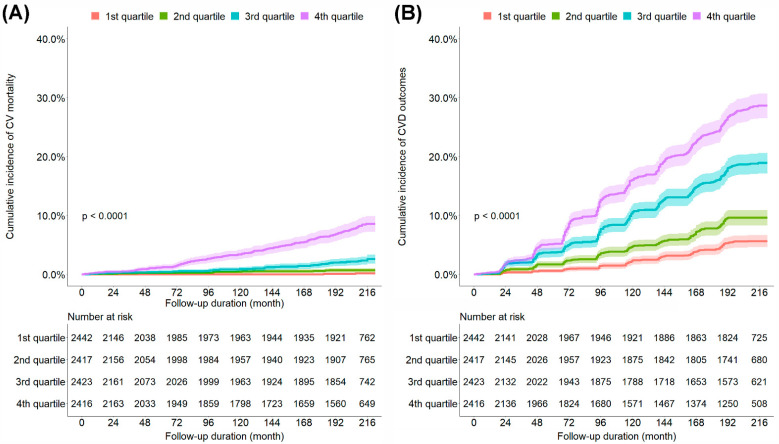
Kaplan–Meier curves comparing the cumulative incidence of CV mortality (**A**) and CVD outcomes (**B**). CV, cardiovascular; CVD, cardiovascular disease; 1st quartile, first quartile of the ePWV; 2nd quartile, second quartile of the ePWV; 3rd quartile, third quartile of the ePWV; 4th quartile, fourth quartile of the ePWV; ePWV, estimated pulse wave velocity.

**Figure 2 jpm-12-01611-f002:**
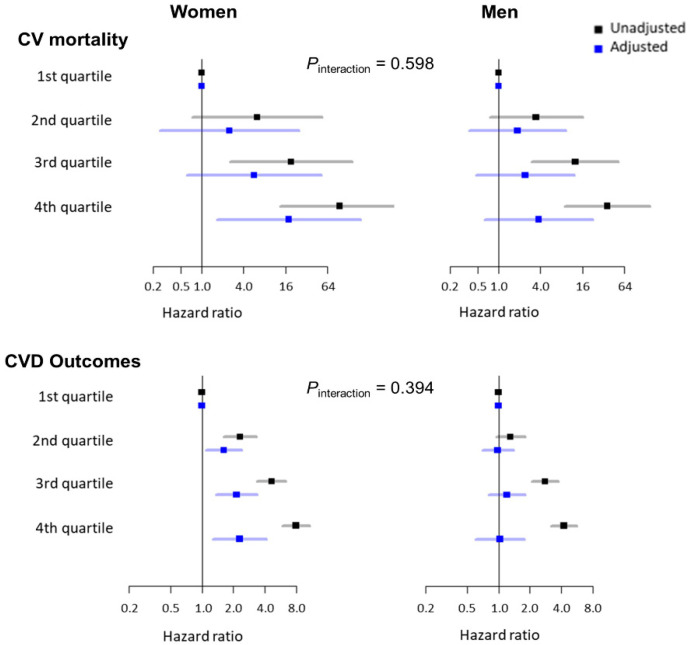
Unadjusted and adjusted association between the ePWV and CV mortality and CVD outcomes according to sex. ePWV, estimated pulse wave velocity; CV, cardiovascular; CVD, cardiovascular disease. Hazard ratios were adjusted for age, systolic blood pressure, smoking status, alcohol drinking status, physical activity, income level, educational status, body mass index, waist circumference, medical history (hypertension, diabetes mellitus, and dyslipidemia), estimated glomerular filtration rate, fasting blood glucose level, total cholesterol level, and low-density lipoprotein cholesterol level.

**Table 1 jpm-12-01611-t001:** Baseline characteristics.

Characteristics	Estimated Pulse Wave Velocity (m/s)				*p*-Value
	First Quartile (4.52–7.38) (*n* = 2442)	Second Quartile (7.39–8.44) (*n* = 2417)	Third Quartile (8.45–9.89) (*n* = 2423)	Fourth Quartile (9.90–15.17) (*n* = 2416)	
Age, mean (SD), year	43.6 (3.2)	47.5 (5.0) *	54.8 (6.6) *†	62.5 (5.34) *†‡	<0.001
Men, *n* (%)	1001 (41.0)	1300 (53.8)	1220 (50.4)	1059 (43.8)	<0.001
Body mass index, mean (SD), kg/m^2^	23.9 (2.9)	24.8 (3.0) *	24.7 (3.2) *	24.7 (3.4) *	<0.001
Waist circumference, mean (SD), cm	79.1 (8.3)	82.7 (8.2) *	84.2 (8.5) *†	85.8 (8.9) *†‡	<0.001
Income level, *n* (%)					<0.001
≥Median	1773 (73.3)	1440 (60.9)	969 (40.7)	504 (21.4)	
Educational status, *n* (%)					<0.001
Lower than middle school	234 (9.6)	482 (20.1)	973 (40.6)	1523 (63.9)	
Middle school	574 (23.6)	632 (26.3)	586 (24.5)	417 (17.5)	
High school	1133 (46.5)	891 (37.1)	594 (24.8)	312 (13.1)	
University and college	494 (20.3)	396 (16.5)	243 (10.1)	133 (5.6)	
Smoking status, *n* (%)					<0.001
Current smoker	670 (27.8)	683 (28.6)	613 (25.6)	501 (21.1)	
Ex-smoker	252 (10.5)	412 (17.3)	404 (16.8)	391 (16.5)	
Never-smoker	1489 (61.8)	1289 (54.1)	1381 (57.6)	1481 (62.4)	
Alcohol drinking status, *n* (%)					<0.001
Current drinker	1229 (50.6)	1282 (53.5)	1106 (46.1)	943 (39.5)	
Ex-drinker	115 (4.7)	143 (6.0)	183 (7.6)	170 (7.1)	
Never-drinker	1085 (44.7)	970 (40.5)	1111 (46.3)	1273 (53.4)	
Physical activity, mean (SD), METs-hour/week	149 (89)	164 (100) *	181 (108) *†	194 (114) *†‡	<0.001
Systolic blood pressure, mean (SD), mmHg	106.6 (8.7)	119.5 (9.7) *	127.8 (13.4) *†	144.3 (17.6) *†‡	<0.001
Diastolic blood pressure, mean (SD), mmHg	70.6 (7.0)	80.6 (7.8) *	84.4 (10.5) *†	91.0 (10.9) *†‡	<0.001
Medical history, *n* (%)					
Hypertension	52 (2.1)	146 (6.0)	428 (17.7)	792 (32.8)	<0.001
Diabetes mellitus	81 (3.3)	95 (3.9)	198 (8.2)	253 (10.5)	<0.001
Dyslipidaemia	52 (2.1)	68 (2.8)	65 (2.7)	43 (1.8)	0.061
Chronic kidney disease	63 (2.6)	54 (2.2)	58 (2.4)	87 (3.6)	0.015
Laboratory data, mean (SD)					
eGFR, mL/min/1.73 m^2^	93.1 (20.2)	90.9 (20.6) *	88.6 (19.7) *†	86.9 (20.4) *†‡	<0.001
Fasting blood glucose level, mg/dL	89.7 (20.5)	91.8 (19.7) *	93.9 (26.4) *†	94.2 (23.9) *†	<0.001
Hemoglobin A1c level, %	5.6 (0.8)	5.7 (0.8) *	5.9 (1.1) *†	6.0 (1.0) *†‡	<0.001
Total cholesterol level, mg/dL	191.7 (34.5)	198.6 (35.6) *	201.9 (37.5) *†	201.4 (38.5) *†	<0.001
Triglyceride level, mg/dL	130.8 (99.3)	152.0 (110.2) *	164.9 (116.0) *†	164.7 (109.3) *†	<0.001
HDL cholesterol level, mg/dL	49.9 (11.5)	49.6 (11.8)	49.2 (11.8)	49.7 (12.4)	0.300
LDL cholesterol level, mg/dL	118.0 (30.4)	122.5 (31.3)	124.5 (32.7)	120.2 (175.7)	0.080

Data are presented as *n* (%) or means (SDs), as appropriate. SD, standard deviation; eGFR, estimated glomerular filtration rate; HDL, high-density lipoprotein; LDL, low-density lipoprotein. * Post hoc *p*: Statistically significant difference *p* < 0.05 compared to the first quartile. † Post hoc *p*: Statistically significant difference *p* < 0.05 compared to the second quartile. ‡ Post hoc *p*: Statistically significant difference *p* < 0.05 compared to the third quartile.

**Table 2 jpm-12-01611-t002:** Hazard ratios for cardiovascular mortality and cardiovascular disease outcomes according to the quartiles of the estimated pulse wave velocity.

Cardiovascular Mortality	Unadjusted HR (95% CI)	Model 1 ^a^ HR (95% CI)	Model 2 ^b^ HR (95% CI)	Model 3 ^c^ HR (95% CI)
First quartile (4.52–7.38 m/s)	REF	REF	REF	REF
Second quartile (7.39–8.44 m/s)	4.68 (1.34–16.27)	3.37 (0.95–11.90)	2.65 (0.74–9.58)	2.11 (0.57–7.84)
Third quartile (8.45–9.89 m/s)	15.80 (4.92–50.75) *	6.01 (1.71–21.19)	4.53 (1.26–16.25)	3.50 (0.95–12.87)
Fourth quartile (9.90–15.17 m/s)	56.60 (18.05–177.43) *†	13.33 (3.45–51.46) *†	10.12 (2.55–40.19) *†	7.57 (1.83–31.25) *†
**Cardiovascular Disease Outcomes ^d^**	**Unadjusted** **HR (95% CI)**	**Model 1 ^a^** **HR (95% CI)**	**Model 2 ^b^** **HR (95% CI)**	**Model 3 ^c^** **HR (95% CI)**
First quartile (4.52–7.38 m/s)	REF	REF	REF	REF
Second quartile (7.39–8.44 m/s)	1.76 (1.39–2.22)	1.36 (1.06–1.74)	1.29 (1.00–1.66)	1.24 (0.96–1.61)
Third quartile (8.45–9.89 m/s)	3.64 (2.95–4.50) *	1.86 (1.41–2.46) *	1.72 (1.29–2.29) *	1.61 (1.19–2.16)
Fourth quartile (9.90–15.17 m/s)	5.85 (4.77–7.18) *†	1.91 (1.33–2.75) *	1.75 (1.20–2.56)	1.56 (1.05–2.31)

HR, hazard ratio; CI, confidence interval; REF, reference. ^a^ Model 1: Adjusted for age (per 10 years), sex, and systolic blood pressure (per 1 mmHg). ^b^ Model 2: Adjusted for variables included in Model 1 and smoking status, alcohol drinking status, physical activity (per 1 METs-hour/week), income level, and educational status. ^c^ Model 3: Adjusted for variables included in Model 2 and body mass index (per 1 kg/m^2^), waist circumference (per 1 cm), medical history (hypertension, diabetes mellitus, and dyslipidaemia), estimated glomerular filtration rate (per 1 mL/min/1.73 m^2^), fasting blood glucose level (per 1 mg/dL), total cholesterol level (per 1 mg/dL), and low-density lipoprotein cholesterol level (per 1 mg/dL). ^d^ Cardiovascular disease outcomes were defined as a composite of cardiovascular mortality, myocardial infarction, coronary artery disease, stroke, heart failure, and peripheral artery disease. * low limit of confidence interval >1 vs. second quartile. † low limit of confidence interval >1 vs. third quartile. The results using second and third quartiles as references are shown in Appendix A.

**Table 3 jpm-12-01611-t003:** Performances and comparisons between Cox proportional hazard models for cardiovascular events.

Cardiovascular Mortality	HR (95% CI)	BIC	ΔBIC ^a^	Harrell’s C-Index	*p*-Value ^b^
Cox models					
10-year ASCVD risk score	1.09 (1.08–1.10)	3688.2	88.9	0.809 (0.784–0.835)	0.008
ePWV (per 1 m/s)	1.87 (1.73–2.02)	3614.9	15.6	0.810 (0.784–0.835)	<0.001
Combined model	1.03 (1.03–1.04) for 10-year ASCVD risk score and 1.29 (1.24–1.35) for ePWV (per 1 m/s)	3599.3		0.824 (0.801–0.849)	
**Cardiovascular Disease Outcomes ^c^**	**HR (95% CI)**	**BIC**	**ΔBIC ^a^**	**Harrell’s C-Index**	***p*-Value ^b^**
Cox models					
10-year ASCVD risk score	1.07 (1.06–1.07)	21,301.8	139.9	0.687 (0.673–0.701)	0.008
ePWV (per 1 m/s)	1.43 (1.39–1.48)	21,207.2	45.3	0.684 (0.669–0.698)	<0.001
Combined model	1.03 (1.03–1.04) for 10-year ASCVD risk score and 1.29 (1.24–1.35) for ePWV (per 1 m/s)	21,161.9		0.697 (0.683–0.711)	

ASCVD, atherosclerotic cardiovascular disease; ePWV, estimated pulse wave velocity; HR, hazard ratio; CI, confidence interval; BIC, Bayesian information criterion; REF, reference. ^a^ ΔBIC was defined difference in BIC between the combined model and other models. ΔBIC < 10 indicates no significant difference between models. ^b^ *p*-value for Harrell’s C-index, which was compared with the combined model. ^c^ Cardiovascular disease outcomes were defined as a composite of cardiovascular mortality, myocardial infarction, coronary artery disease, stroke, heart failure, and peripheral artery disease.

## Data Availability

The the KoGES data are available on reasonable request from the Korea Center for Disease Control and Prevention website (http://nih.go.kr/contents.es?mid=a40504060100, accessed on 30 November 2021).

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
