# Peer review of "Association of the Estimated Pulse Wave Velocity with Cardio-Vascular Disease Outcomes among Men and Women Aged 40–69 Years in the Korean Population: An 18-Year Follow-Up Report on the Ansung–Ansan Cohort in the Korean Genome Environment Study"

_jpm, 2022, doi:10.3390/jpm12101611_

Round 1
Reviewer 1 Report
Authors are to be congratulated to their excellent work.
The author conducted a a prospective study from the Ansung-Ansan cohort, which consisting of 10,030 South Koreans aged 40-69 years residing in two cities, Ansung and Ansan. The study was well conducted and the manuscript has a good structure, results are clear, and conclusions are supported by data. Findings are plausible as consistent with previous Vlachopoulos's study (PMID: 31596491).This association was more clearly observed among the women than among the men. An elevated ePWV is associated with CV mortality and CVD outcomes. The ePWV is expected to serve as a potential marker for identifying high-risk groups for CVD events.However, there are some issues that should be addressed.
1.The best cut-off of ePWV level for discrimination between patients with and without cardiovascular disease outcome.
2.However, can you provide medication or associated treatment, because of treatment often have positive or negative influence on ePWV.
3.Do you validate your ePWV with PWV in your cohort.
4. You evaluate ePWV and cardiovascular events according to sex, however, regarding DM, hypertension or CKD, is it the similar finding?
Reviewer 2 Report
In this work, Kim and colleagues want to evaluate if estimated pulse wave velocity (ePWV) could predict adverse cardiovascular disease (CVD) outcomes in subjects with increased CVD risks in the general population. Ten thousand thirty patients aged 40– 69 years from the Ansung–Ansan cohort in a prospective community-based cohort study were followed up for over 18 years. In a population of 9698 participants it was demonstrated that an elevated ePWV was associated with CV mortality and CVD outcomes. The paper is clear, straightforward, and documented.
Questions:
1. Page 3, line 118: it is not clear the definition of “CAD other than IMA...” It must be reformulated.
2. Page 5, Table 1: it shows baseline characteristics of the partecipants. It is reported the ANOVA analysis, but it is not clear if there are statistically significant differences between the various quartiles.
3. Page 6, Table 2: it shows Hazard ratios for cardiovascular mortality and cardiovascular disease outcomes according to the quartiles of the estimated pulse wave velocity, but also in this case but it is not clear if there are statistically significant differences between the various quartiles.
4. Page 7, line 232 there is an open parenthesis without closure, something is missing?
5. Page 7, Figure 2: after adjustments for relevant variables (among which age stands out), not only there were no significant differences in the risk of CV mortality and CVD out-comes according to the ePWV quartiles among the men, but also the rising trend is lost in women. How do you explain it? There may be a role in the fact that ePWV was calculated from the age of the patients themselves?
6. Page 8, Table 4: It is not clear if Harrell’s C-index significantly increases in the 3 models, it must be reformulated.
Round 2
Reviewer 1 Report
Authors are to be congratulated to their excellent work and I do not have other comments